# Retirement Rhythms: Retirees' Management of Time and Activities in Denmark

**Aske Juul Lassen [1],***, **Kenneth Mertz [2]**, **Lars Holm [3]** 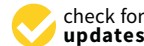 **and Astrid Pernille Jespersen [1]**

[1]   Center for Health Research in the Humanities, Saxo_Institute, University of Copenhagen,
     2300 Copenhagen, Denmark; apj@hum.ku.dk
[2]   Institute of Sports Medicine, Department of Orthopaedic Surgery M, Bispebjerg Hospital,
     2400 Copenhagen, Denmark; kenneth.mertz@sund.ku.dk
[3]   School of Sport, Exercise and Rehabilitation Sciences, University of Birmingham, Edgbaston,
     Birmingham B15 2TT, UK; L.Holm@bham.ac.uk
*   Correspondence: ajlas@hum.ku.dk

**Abstract:** We scrutinize how the everyday lives of well-educated and healthy Danish retirees are structured and experienced and study how they organise their days and weeks. Our aim is to investigate how retirees manage and organise time and the ways these relate to societal rhythms in order to contribute to theories of retirement and social gerontology. We have combined qualitative (individual interviews, focus group interviews, design games, and drawings) and quantitative (activity monitoring, sleep quality, and health markers) data from 13 participants over the age of 65 years, who are research participants in a randomised controlled trial (RCT). Our interdisciplinary dataset allows us to analyse and compare subjective experiences of everyday activities with objective measures of daily activities. The older adults lead busy lives with many diverse activities and use these to structure their everyday lives in ways resembling the rhythms of the labour market with organised and busy mornings and loose afternoons and evenings. We discuss how our findings relate to continuity theory and suggest that Lefebvre's rhythmanalysis allows us to study the retirement rhythms of older adults as part of both biological, social, and societal rhythms. This has practical as well as conceptual implications.

**Keywords:** theories of retirement; rhythmanalysis; management of time; interdisciplinary gerontology; busy ethic; societal rhythms

## 1. Introduction

Since the proliferation of retirement in the mid 20th century, social gerontology has shared an interest in the role of time in the everyday life of older people, e.g., [1,2]. This led to an interest in the proper leisure activities older people should pursue in order to achieve a good old age, e.g., [3,4], and not become idle due to so-called 'time in abundance' [5] (p. 20). In recent decades, this rather pessimistic view on the 'time in abundance' has been reversed to an overall discourse formulating freedom in old age, i.e., the freedom to engage in the leisure activities that one wants [6]. However, as Ekerdt and Koss have described, retirees do not just approach time as 'free' but tend to 'organise activities into recurring sequences, into routines' [7] (p. 1297). This routinization of activities is a structuring strategy in retirement enabling retirees to stick to their exercise patterns, social engagements, medical regimens, and other frequent habits of daily living. Routinization of activities is a way to handle the open-endedness—both embedded in-time in abundance' and in 'freedom'—of retirement and to establish continuity from work-life.

In this paper, we elaborate on Ekerdt and Koss' argument by adding that the routinization of activities and time management is not only an individual pursuit and identity-making. It is also embedded in collective and societal expectations and practices, and as such part and parcel of contemporary society. We scrutinize how the everyday lives of a group of Danish retirees are structured and experienced and pose the research question: how do the new generation of active, healthy, and well-educated Danish older people manage time and organise their days and weeks? Our hypothesis is, that while retirement may lead to changes in the management of time, we will also find a continuity of rhythms in everyday life post-retirement.

We suggest that this study of retirees' everyday lives is timely. Retirement is changing in Denmark, as well as in the Western world in general. In 1892, when Denmark saw its first state-sponsored old age benefits, few people would reach retirement age and meet the criteria for obtaining pensions [8]. In 1956, universal state pension became a right in Denmark from the age of 67, and through the latter half of the 20th century, retirement age was decreasing, and early retirement proliferated. In recent years, retirement age has increased again [9], and important transformations of the management and organisation of retirement are taking place in the Western world [10]. In Denmark, the age at which citizens can obtain pensions is rising continuously in the coming decades (in 2020 it is 66 years of age), but at the same time an increasing percentage of workers work beyond retirement age. In 2017, 18 per cent of the 67-year olds were employed [11]. Many engage in gradual retirement, and entrepreneurship around retirement age is widespread [12]. Moreover, the citizens that do retire engage in volunteer work, exercise, as well as cultural and political activities [13–15]. As such, our study is both culturally rooted in Denmark, as well as pointing to some overall tendencies regarding retirement in the Western world.

Furthermore, we suggest that in order to portray retirement elaborately, we need an interdisciplinary approach with a range of different ways of measuring and probing. We know from a range of social science studies that retirees portray themselves as active (e.g., [13–16]). While we do not find any reason to doubt that this is so, we need to examine what they mean with being active—what is it they do, and how often? When they say they are more active during the week than in the weekend, is this visible when measuring step counts, or is the increased activity level different than what can be measured? As such, our combination of methods enables us to dissect that a step is not just a step. There are qualitative differences between steps at different times and during different kinds of activities.

By combining qualitative and quantitative data from thirteen men and women over the age of 65 years, we probe into the routinized everyday lives of retirees, their activities, and their daily and weekly rhythms. Our aim is to investigate how people manage time, routines, and continuity when retired. Our diverse and rare dataset allows us to analyse and compare subjective experiences of everyday activities with objective measures of daily activities. Following the call from Henkens and colleagues [17], we probe into the nature of contemporary retirement and contribute to social theories of retirement. Many studies describe contemporary retirement through discourses of active and healthy ageing [18–21], but hitherto such studies have remained largely conceptual, while empirical studies of active ageing tend to critique the active or successful ageing agenda [22,23]. We propose that our close-up description and analysis of everyday life in contemporary retirement is much-needed, as we scrutinise the routinised retirement rhythms of active and healthy retirees.

As historian Thomas Cole shows, in the latter half of the 19th century, old age fell victim to an increasing focus on production [24] (pp. 161–190). When the human being was reduced to its productive capacities, old people became obstacles to industrial progress. This in part has led to a 'busy ethic', which extends the busy life into old age by endorsing an active life [25]. Through such a moral continuity from work-life, the 'good old age' has become synonymous to the 'busy old age' [26], forming a 'new gerontology' aspiring to form successful aging as an individual endeavour [27], with resigning 'villains of old age' [28] or on the contrary 'exemplars of retirement' [29] working after retirement age. Moreover, through a variety of scientific discoveries stressing the benefits of an active lifestyle in old age, policy programs of active ageing have become scientifically legitimised [30]. When the good old age is the active and busy old age, time is no longer in abundance, but a valuable resource.

This resource can be analytically explored in multiple ways. Although Birren stated that (chronological) time is a hollow independent variable [31], the main body of literature in gerontology dedicated to time focuses on time as linear, by focusing on age as chronological, or as one of the many alternative age-qualifications such as biological, functional, or mental age. Ideas of generations [32], age norms [33], or age stratification [34] all pose linear time and the way it is woven into our social lives [35] as their premises. As Baars and colleagues have put it, chronological time is a 'problematic foundation' for gerontology [36] that creates chronological regimes [37] decisive for participation in the labour market or the access to social services. Others suggest that time is something people actively engage with and appropriate in their everyday lives, in a kind of 'time work' [38].

Based on our data, we suggest a rhythmic approach to time inspired by the French tradition of rhythmanalysis proposed by Henri Lefebvre [39], which centres on the ways different rhythms (biological, social, individual, societal) relate to one another, impose on one another, and become partly fused. From this perspective, retirement suggests both a different rhythm in the everyday life of people, but also continuity of rhythms from work-life. In one of their essays on rhythmanalysis, Lefebvre and Catherine Regulier show how the way we measure time, indeed the notion of everyday life, is modelled over an abstract, quantitative work-time that stems back to the introduction of the industrial labour force. Since the industrial revolution, the clock has formed everyday life, sleeping, meals, relations, and families [40]. But this clock-rhythm is at the same time traversed by biological rhythms (diurnal clock, heart, nerves, gut, etc.) and cosmic rhythms (day, night, seasons). The rhythms interact and interlock in such a way that we become hungry when the clock says it is mealtime. Such deep integration between different time measures and biological and social rhythms is not likely to suddenly cease due to retirement. The retirement rhythms may be so deeply affected by general societal rhythms (often centred on work and consumption) that changing them is slow and tedious.

With this rhythm-analytical approach, we are able to show how contemporary retirement is intrinsically part of the rhythms of society. Retirement is not solely an individualised life phase, but also part of contemporary society with rhythms grounded in the labour market and its busy ethics.

## 2. Materials and Methods

The paper is based on data from the Danish research project Counteracting Age-Related Loss of Skeletal Muscle Mass (CALM), which investigates how to counteract age-related loss of skeletal muscle mass before the onset of age-related diseases. The first part of CALM was organised around a randomised controlled trial (RCT) wherein 208 research participants over the age of 65 years were randomised into 5 groups and were enrolled through a 12-month trial, where they were subjects to physiological and microbiological tests, as well as ethnological analyses [41]. Each group differed in terms of training regimen and protein supply in order to study the effects of these factors on their muscle mass. Participants were recruited through newspapers, social media, and networking at senior centres. There were 18 exclusion criteria, which ensured that we only included relatively healthy participants, who at the same time did not exercise vigorously (see [41] for a detailed description). As we show below, it is solely parts of the physiological (quantitative) and ethnological (qualitative) material we have used for this paper.

The second part of CALM was organised around an innovation project. At this point, the knowledge gained from the trial about muscle mass, eating, and exercise habits was challenged and tested in a range of experimental workshops with retirees, ethnological and historical analyses of physical activity, daily routines and habits as well as food protein and sensory tests of a range of protein-rich food products. As such, we tested the data in order to see whether, for example, the best options of counteracting muscle mass (i.e., specific eating or exercise regimens) also would be translatable to the everyday practices of the target groups (people 65+). Thus, the innovative aspect of the trial was to see if the ideal solution in the lab would also function in the everyday practices of the target group.

These many types of data provide new insights and different stories of contemporary retirement. As such, we use these data to explore how the target population manage time in their day-to-day life. In this paper, we analyse 13 of the participants who volunteered for both the RCT as well as the experimental workshops. All participant names have been changed in order to ensure anonymity. However, we have obtained informed consent to use pictures and drawings from the participants, and, hence, have chosen to use participant #1's drawing of her daily schedule (see Figure 1).

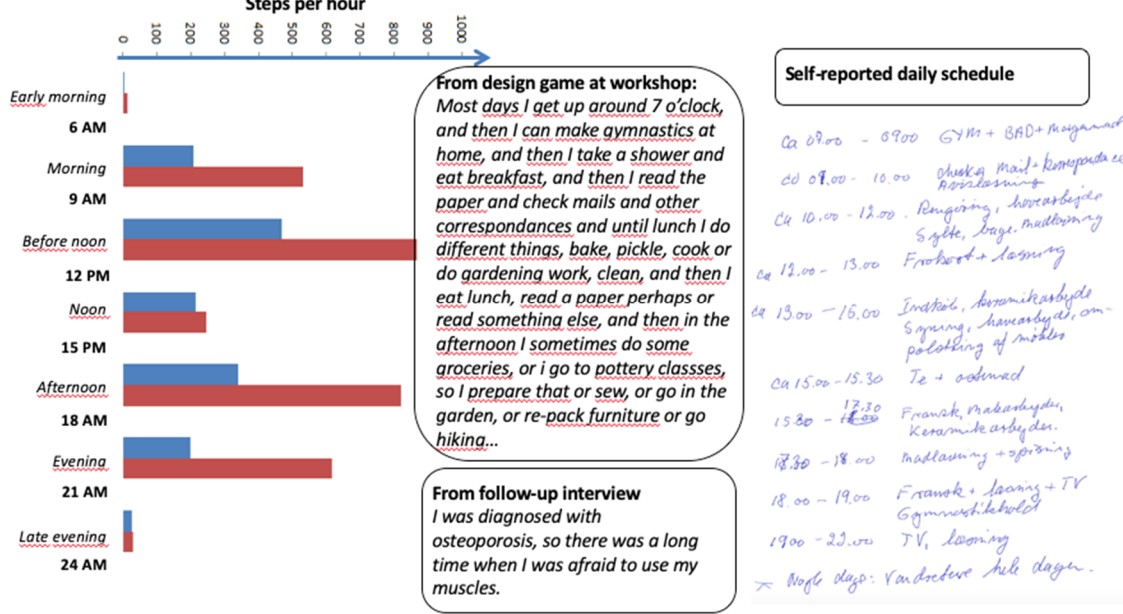

**Figure 1.** Participant #1: Woman, 68 years of age. A retired teacher (among many other things), who retired just before entering the project. To keep active, she had 3 cleaning jobs, but during the trial she was diagnosed with osteoporosis and quit her jobs. She gets up early and does gymnastics to the radio in her home 3 times weekly. She has a very busy schedule throughout the day.

The data for this paper consists of:

Qualitative data:

Screening and follow-up interviews: The 208 participants in the RCT were interviewed at baseline and at follow-up 18 months later. The interviews lasted between 5–15 min, followed standardized interview guides—one for the baseline and one for the follow-up—and were conducted by the trial staff as part of the test days. In the screening interviews, participants were asked about exercise and eating habits, living arrangements, reasons for participating in the trial, etc. In the follow-up interviews, participants were asked whether they had changed habits during the trial, and how they had experienced their participation in the trial. We followed this procedure to get some qualitative data, since it was not possible to conduct in-depth interviews with all participants in a sample of this size.

Focus group interviews, design games, and drawing of daily schedules:

Six experimental workshops were conducted with 84 participants in total. Three focused on eating practices and three focused on physical activity practices. One workshop from each theme had participants from the RCT. We chose this setup, as most of the trial participants (*n* = 208) were from the Greater Copenhagen area, and we wanted a broader sample for the experimental workshops. As such, it was only the two workshops conducted in Copenhagen that included trial participants. For this paper, we are using data from the workshop about physical activity with CALM participants conducted in November 2016 (*n* = 13). At the workshop, we conducted a focus group in which we divided the participants in three groups and discussed everyday routines and the ways physical activity was part

of their everyday lives. As the workshop on physical activity included most details on everyday life rhythms, we have chosen to only include data from this workshop in this article.

We also played a design game made by the researchers conducting the workshop. The participants were divided into three different groups, and they played out different physical activity situations.

At the same workshop, we asked participants to draw a timeline of a typical day in their lives. Most responded that each day was different from the next, and hence, that they would not be able to draw only one timeline. However, when drawing, they all had some daily routines they drew and/or wrote (see Figure 1). Thus, while the drawn timelines do not encompass all their activities, they show daily patterns and habits.

All qualitative data have been transcribed verbatim and coded and thematized using an inductive approach [42] in which major underlying patterns and themes have been identified. The initial coding sessions involved eight persons (authors, student assistants, interns, and colleagues). Later in the process, the coding was conducted and discussed by authors A.J.L. and A.P.J.

Quantitative data:

Activity monitoring: Daily activity levels were measured by mounting an accelerometer-based activity monitor (activPal 3™, activPal 3c™, or activPal micro; PAL technologies, Glasgow, UK) on the anterior surface of the thigh [43]. The monitor was worn for 96 continuous hours covering a full weekend. Data are represented as average steps per hour during 3 h intervals as well as total steps per day.

Sleep quality:

Sleep quality was assessed using the Pittsburgh Sleep Quality Index (PSQI) [44]. The sum of seven component scores provides a global score, with a higher score being reflective of poorer sleep quality. A global score greater than 5 is indicative of poor sleep quality [44]. As a part of the questionnaire, participants noted the time of day they went to bed, and time of day they got up in the morning. Besides bedtimes highlighting the daily rhythm of participants, the sleep quality assessment was mainly used to characterize the sample.

Health markers:

As a part of the clinical trial, the height and weight of the participants were measured and reported as body mass index (body weight [kg] × height$^{-2}$ [m], BMI. Blood pressure was measured in the resting condition. Handgrip strength of the dominant hand was assessed using a grip dynamometer (DHD-1 [SH1001]; SAEHAN Corporation, Changwon City, South Korea). These markers were mainly used to characterise the sample.

With this combination of qualitative and quantitative data, we are able to provide a detailed and nuanced picture of the everyday lives of Danish retirees and to show how the biological, the physiological, and the social are entangled in everyday life. We collected 1000+ measures from 50+ measurement techniques on each participant ranging from gender, education, employment, and civil status to food registration, Short-Form 36, muscle biopsies, and gut microbiomes (for a more detailed description, see [41]). However, as we only focus on 13 participants, the results in this article is of a qualitative nature, and the quantitative data are mainly used to discuss the qualitative findings and characterise the sample.

## 3. Results and Analysis

In Table 1, we have gathered a range of data from the participants, in order to provide some context on the kind of subjects from whom we developed our results and analysis. As Table 1 indicates, the participants were well-functioning and healthy (based on the parameters measured here). For the women, BMI was between 20 and 25 kg/m$^2$ and for the men, BMI was slightly higher but below 28 kg/m$^2$. Blood pressure was on average within recommended range with one woman above in

systolic pressure and a few men slightly higher. According to the measured steps per day, most participants could be categorized as active or highly active (<10,000 steps per day), although two women could be categorized as low active (<7500 steps per day) or sedentary (<5000 steps per day). For both women and men, grip strength was higher than cut points for sarcopenia, and sleep quality was good for all but one of each sex. Moreover, eight of the 13 participants engaged in either paid or voluntary work. All but one had at least three years of higher education (one did not respond). All participants lived within commuting distance of Copenhagen, with three participants living in the city, eight in the suburbs, and two in the countryside.

Our data portray the everyday life of a sub-group of the older population with a high level of education, income, and health. While this group is characteristic of a new generation of retirees in Westernized cultures, they are not representative of the retired Danish population. In Denmark, roughly one quarter of the 65+ group have three or more years of education after high school [45], while all participants in our study have at least three years. Roughly 17 percent of retirees in Denmark only have state pension and are relatively poor [46], while all participants in the study have stable economies. In addition, especially the female participants have lower BMIs than average for their age group, as 78 per cent of Danish women 65–74 y/o have BMI of 25 or more [45]. Six participants (46 per cent) engage in volunteer work, which is representative for the Danish population aged 65–69 of which 40 per cent engage in volunteer work [47].

While they form part of a privileged group—also in a Danish context of the welfare state—they also carry many traits typical for their generation embodying the message of active and healthy ageing (as well as containing a 46 per cent (*n* = 6) degree of divorcees). A close-up and nuanced description of this group's everyday lives and their management of time has been rarely investigated in the current literature.

*3.1. Participant #1*

In the following, we will use research participant #1 to exemplify the everyday lives we have seen portrayed and lived among participants (see Figure 1). #1 described her life as busy and had many activities. While #1 engaged in more activities than other participants, her way of describing her day resembled the other participants. While #1 stayed more in her house than average among participants, which she explained is due to her osteoporosis, she and the others used activities to structure their everyday lives in the absence of an institutionalised work-schedule. When asked about their everyday life, they responded with listing the activities (exercise, volunteering, classes, hobbies, grandchildren) that they engaged in and how other activities (meals, daily chores, mails, etc.) fit into these activities. All participants described a kind of physical activity (walking, biking, strength training, running, swimming, etc.), which usually took place between breakfast and lunch, but those that attended classes (yoga, gymnastics, etc.) often did so in the afternoon or at night.

#1 slept 7 h per night, got up between 6 and 7 AM, and scores 2 on the Pittsburgh sleep quality index, and all participants except two reported very high sleep quality. Of the 12 participants reporting sleep, three reported to rise later than 8 AM. They would usually get up at the same time during the week and prepare for the day, as they had done while working. However, we must note here, that we have merely collected data about their previous rhythms retrospectively through the interviews. For many, this continuity entailed taking a bath and then preparing and eating breakfast. As #1, participants would generally start the chores of the day after breakfast, although three participants (#6, 7 and 10) chose to postpone the chores and take a bike-ride or go hiking. They described their mornings as busy and packed with assignments, such as gardening, cleaning, doing laundry, doing groceries, and exercising. Like #1, after lunch, participants' days were less structured, with time for reading books, meeting friends, having coffee, etc. Similar to #1, the evenings were often spent in front of the television or reading, if they did not report going out or seeing friends. While all participants reported watching television at night-time, they all refrained from daytime TV-watching or napping (with the exception of #5 and #10 who reported napping as an integral part of their daily routines after lunch).

**Table 1.** Characteristics of participants.

| ID no. | Sex | Age [Years] | Retirement Age [Years] | (Former) Occupation | Education | Work Security/ Stability | (Volunteer) Work | Marital Status | Sleep Quality Index | BMI [kg/m$^2$] | Blood Pressure Syst/Diast [mm Hg] | Grip Strength [kg] | Daily Step Count | | Kinds of PA | App. Bedtimes | |
|---|---|---|---|---|---|---|---|---|---|---|---|---|---|---|---|---|---|
| | | | | | | | | | | | | | Weekdays | Weekend | | Wake up | Go to Bed |
| 1 | F | 68 | 63 | Telegraphist, teacher | Telegraphist, teacher | Stable | Cleaning 5 times weekly, paid | Divorced | 2 | 22.3 | 106/69 | 26.5 | 5308 | 9404 | Gymnastics, hiking, cleaning | 7:00 | Not specified |
| 2 | M | 66 | 63 | Public administr-ration | Office education | Stable | cleaning at restaurant as volunteer, board member in social organizations | Divorced | 1 | 23.4 | 117/82 | 51.4 | 10,684 | 8993 | Biking, winter bathing | 8:00 | 0:00 |
| 3 | F | 67 | 68 | Head of public school | Teacher | Stable | Board member in the apartment complex | Divorced | 9 | 22.3 | 143/75 | 31.7 | 5348 | 5380 | Swimming, biking | 7:00 | Not specified |
| 4 | F | 71 | 70 | Consultant | Process technologist | Stable | At NGOs | Divorced | 1 | 21.5 | 97/72 | 25.3 | 9720 | 14,654 | Morning gymnastics, biking, running | 7:00 | 0:00 |
| 5 | M | 70 | - | Many small jobs | Historian | Unstable | None | Married | 3 | 22.9 | 143/89 | 41.9 | 11,603 | 9594 | Walking | 6:00 | 23:00 |
| 6 | M | 66 | 65 | Pedagogue | Pedagogue | Stable | None | Married | 3 | 26.8 | 153/83 | 47.1 | 16,562 | 10,129 | Biking, swimming, yoga | 5:00 | 23:00 |
| 7 | M | 70 | 67 | High school teacher | High school teacher | Stable | Fills in and plays the organ in a church sometimes | Divorced | 7 | 23.5 | 120/66 | 36.8 | 13,509 | 10,225 | Morning gymnastics, running | 8:00 | Not specified |
| 8 | F | 71 | 67 | Teacher | Teacher | Stable | Sensory examiner, paid | Divorce-d | 1 | 20.4 | 177/73 | 35.8 | 13,234 | 11,526 | Running, walking, golf | Varies | |

**Table 1.** *Cont.*

| ID no. | Sex | Age [Years] | Retirement Age [Years] | (Former) Occupation | Education | Work Security/ Stability | (Volunteer) Work | Marital Status | Sleep Quality Index | BMI [kg/m$^2$] | Blood Pressure Syst/Diast [mm Hg] | Grip Strength [kg] | Daily Step Count | | Kinds of PA | App. Bedtimes | |
|---|---|---|---|---|---|---|---|---|---|---|---|---|---|---|---|---|---|
| | | | | | | | | | | | | | Weekdays | Weekend | | Wake up | Go to Bed |
| 9 | M | 70 | | | | Stable | None | Married | 2 | 22.5 | 160/75 | 52.4 | 9862 | 11,697 | Athletics five times a week—gym, running or other kinds of exercise | 7:00 | 22:30 |
| 10 | M | 68 | 65 | Pedagogue | Pedagogue | Stable | None | Married | 2 | 27.5 | 183/89 | 35.9 | 7594 | 14,653 | Running/ power-walking, biking | 7:00 | 23:00 |
| 11 | M | 73 | | Public administer-ration | Accountant | Stable | Volunteer handyman in DaneAge (association for older people) | Married | 2 | 28.0 | 168/96 | 43.3 | 13,840 | 11,596 | Home exercises | 7:30 | 23:00 |
| 12 | F | 68 | 66 | | | Stable | Volunteer in a knitting café | Married | 2 | 22.0 | 139/84 | 36.8 | 10,438 | 11,141 | Hiking, gymnastics, walking | 8:00 | 0:00 |
| 13 | F | 67 | | | | | | Married | - | 24.7 | 133/86 | 22.1 | 14,011 | 11,842 | Walking, weightlifting for elders, yoga | 7:00 | Not specified |

Mean = 68.846; SD = 2.1543; sleep quality index: good sleep quality; ≤5 points, poor sleep quality; >5 points; BMI: normal weight; 20–25 kg/m$^2$, overweight; 25–30 kg/m$^2$; blood pressure: <140/90 mm Hg; grip strength: men; 40 kg, women; 25 kg (Hansen, Beyer, Flensborg-Madsen, Grønbæk, & Helge, 2013); step count: 7500 steps/day, active; 10,000–12,499 steps/day, highly active; >12,500 steps/day (Tudor-Locke & Bassett, 2004).

In the qualitative data, weekends and vacations continued to have the same meaning for the participants in the study. For #1, the weekends were considered 'time off' and many participants would stay up late, see friends, or go out on Fridays and Saturdays, as well as sleep late on Sundays.

However, for #1 and for participants in general, this distinction could not be seen on the step count data obtained with ActivPal. Participants got out of bed at the same time at the weekends and weekdays and achieved more or less the same number of daily steps.

In the following, we will unfold these relations between the different kinds of data and discuss some possible explanations behind our findings.

### 3.2. Rhythms of Everyday Life

We find numerous cyclical repetitions in the observations about the older adults' lives. We also find that the structure of the day during the week in many ways resembles a workday. The rhythms of work-life persist, both daily in terms of wake-up-time, mealtimes, and commencing activities, and weekly, in terms of a sharp differentiation between weekdays and weekends. Our data support the idea that the subjective meanings of work shift with retirement [48]. To this end, the retirees in this study commence their 'work-day' after breakfast by doing chores such as groceries, garden work, cleaning up, washing clothes, checking mail, and exercising. Lunch constitutes the end of the 'short' workday for many of the participants, although some engage in volunteer work in the afternoon. After lunch, the days tend to be much more varied and open-ended:

> In the afternoon, I don't have the same kind of schedule. I do things, but more like puttering around, reading, knitting, meeting friends at cafés, having them over, walking or the like. On Tuesdays I pick up the grandchildren from school and kindergarten. (#12 during focus group).

In line with #12, other participants describe their activities in the afternoon as puttering around, when they are not picking up grandchildren from kindergarten or school or attending planned activities. For those participants where the grandchildren live far away, picking up grandchildren did not form part of the weekly rhythm. However, for many of the participants living close to their grandchildren, they are an integral part of the weekly rhythm.

This continued work-rhythm could be explained in various ways.

In gerontology, the continuity of activities and interests is a well-described adaptive strategy within the framework of continuity theory, e.g., [49,50]. In this line of thought, adaptation is seen as a strategic way to maintain continuity of identity by linking change to individual life history (internal continuity) and physical and social structures (external continuity). In both cases, adaptation is an assessment (sometimes unconscious) of how the present links to the past [49].

However, from a rhythmanalytical perspective, the continuity theory is too focused on individual assessments and linear time. Regarding the individual assessment, the societal rhythms proceed unnoticed by individuals' retirement, but the retiree continues to take part in and reproduce societal rhythms. As such, retirement is not just an individual but also a societal endeavour. The participants in the study do not continue their work-rhythms as an individual choice. Rather, they form part of a societal rhythm, and activities and mealtimes are structuring components in their daily lives.

### 3.3. Linear and Cyclical Time

In rhythmanalysis, linear time is merely one version of time with cyclical time being just as important for social life. While linear time is the years passing and never returning, cyclical time is the yearly repetition of spring, summer, fall, winter, spring, etc. Therefore, rhythms are cyclical—from traffic lights continuously repeating the same cycle, to rush hours, public transport schedules, shops' opening hours, etc.—and do not suggest linear progression or sudden change. These rhythms continue despite retirement and are related to the rhythms of the labour market and society. This relation between the cyclical and the linear is what Lefebvre pointed to as the interference between days,

seasons etc. (cyclical repetitive) and practice in the form of work or retirement tasks (linear repetitive). There is a cycle onto which a linear practice is performed. The clock is an example of such cyclical repetition accompanied by a linear tick-tock, and it is precisely through this relation between the cyclical and the linear that we can measure and organise time [39] (p. 18).

The importance of societal rhythms can be illustrated by the importance of the weekly cycle in our data. Unlike days, months or years—which stem from a universal rhythm—the week is a social convention with deep roots in social structures like religion and the institutionalisation of labour [51]. Despite the immediate insignificance of weekdays and weekends in retirement, the seven-day circle continues to play a huge role. While weeks tend to lose their meaning and are unmasked as social conventions for soldiers in war, new mothers, or students preparing for an exam [51] (p. 138), this was not the case for the retirees in our study. As described above, the qualitative data suggested clear differences between weekends and weekdays in time of getting up in the morning (later in weekends) social activities at night-time (more activity at Friday and Saturday nights) and levels of physical activity (more physical activity during weekdays):

> I must admit I tend to see a lot of sports in the tele during the weekends. Exercise does not really fit into my scheme. It suits me better during the week. (#2 during focus group)

This difference between weekends and weekdays was evident in much of the participants' talks about their drawings of daily schedules. When prompting them to draw, we asked for a time-line of a typical day in their lives, and all the activities drawn and enlisted were weekday activities. The drawings urged participants to talk about time both as linear progression (from morning to night), linear repetitive (typical assignments during the day) and as cyclical repetitive (weekly rhythms). As such, the researchers also inclined participants to think of time and assignments in a rhythmanalytical manner. What became evident in the subsequent interviews was that the participants used activities, mealtimes, and other routines to structure everyday life, as also described by Ekerdt and Koss [7]. As such, time and assignments are interwoven and rhythmsanalysis enables us to describe how time is woven into collective rhythms.

Likewise, many participants described seasonal rhythms, and the passing of the year structured activities like hiking and cycling. Denmark is far north with long days during summer (approximately 18 h) and short days during winter (approximately 7 h). Some participants lived in their allotment yards during the summer. In addition, many of the activities listed by participants were closed during the summer months, which caused participants to get in bad shape, as they were not able to maintain their exercise routines without the structure of classes and social commitments. Thus, the yearly rhythm influenced activities both in structural (closed classes during summer) and personal (not running or walking during winter) ways.

As described, the difference between weekdays and weekends was not visible in the activity registration data (see Table 1). This, we ascribe to the fact that the ActivPal device is an accelerometer, measuring activity based upon changes in position, and categorizing these recordings into sitting/lying, standing, or walking. Therefore, the objectively similar activity patterns between weekdays and weekends can be ascribed to the fact that the ActivPal does not distinguish between types of movements. Further, we have summed up the activity registrations in 3 h slots throughout the day. Whether frequent and fast steps are taken within, e.g., one hour, and sitting time makes up the remaining 2 h time or, less frequently, more calm steps are spread over the 3 h slot, which results in the same ActivPal activity registration. Therefore, in relation to the activities performed, the ActivPal cannot distinguish between a weekday morning with house cleaning and grocery shopping and a weekend day playing with grandchildren at a playground. Thus, the ActivPal is an objective measure of movement and cannot detect the purpose and intention of the movement, nor how the older adults perceive the movement, or the feelings related to it. Therefore, the difference between weekdays and weekends is caused by an experienced difference amongst participants, which is not reflected when measured.

### 3.4. Sleeping Patterns: Fused Societal and Biological Rhythms

Wake up time is, again from a rhythmanalytical perspective, an illustrative example of the ways societal rhythms are fused with the biological rhythms of retirees. The rhythms are socio/bio/cultural. The high scores in sleeping quality can be partly explained by participants' high levels of physical activity. Previous research suggests that physical activity interventions improves the subjective quality of sleep [52] and, oppositely, older adults with poor sleep quality and short sleep duration are less likely to be sufficiently physically active [53]. While there is a biological need to sleep for a certain number of hours per day, the overall continuity of wake up time from work-life to retirement also suggests that the wake up time conditioned by the labour market during work-life has become such a deep rhythm that it continues post-retirement. While a few participants suggest that they have a harder time sleeping until late than previously, others wake up at the same time as always, while others continue to set the alarm clock on weekdays.

Just like the stomach says it is hungry at mealtimes, so do participants wake up (or enforce to be woken up by the alarm) at the same time as they have done during work-life. With this, we do not attempt to establish an exclusive social or biological explanation for participants' wake up times or tendency to have busy mornings. Rather, we suggest that the busy mornings are both explained by a societal rhythm continued from work, as well as other biosocial phenomena, such as participants stating that they 'are more fresh in the beginning of the day' (#12), or that they 'need to get the day started to get anything done' (#4), or that 'you need to work before you can enjoy' (#6).

### 3.5. Chrononormativity

As described above, there are certain times allocated for different activities during participants' daily, weekly, and annual rhythms: Physical activity times, shopping times, gardening times, mealtimes, sleeping times, volunteering times, etc. Participants distinguish between mornings, where the activities resemble work in the form of chores and physical activities, and afternoons, where they 'putter around' and visit friends:

> We get up around 7AM and pick up the newspaper, and sit and read it, and then around 8 o'clock we are doing our normal chores, house and garden and such, if I am not doing gymnastics at the club. I exercise 5–6 times a week, but sometimes at night. It can be weightlifting, running or technique. And then we do groceries after our morning training. My wife also does gymnastics, so we often do our exercise together. After lunch we read a lot and putter around or visit friends. Sometimes we pick up the grandchildren. (#9 during design game)

While work changes character from pre- to post-retirement, there is still work to be done [48]. This concept of work is in line with the 'busy ethics' [25]. In the participants' statements about work and busyness, they also reveal a normative stand towards the ideal retiree. The participants performed this ideal in our data and talked about themselves as being busy. The accelerometers also revealed high levels of physical activity, but besides this, our methods only allowed us to collect accounts about their behaviour. However, it is important to note that busyness is just as much a sensation as an objective fact. While our data cannot prove how busy these participants actually are, it reveals both that they feel that they are busy, as well as feel a need to display themselves as being busy retirees.

These ideal of busyness also partly explains why there is a range of taboo activities during daytime. Although napping has been suggested to be beneficial to improve mental and physical health [54], many participants describe it as socially unacceptable to take naps or watch television during daytime. Similar findings regarding napping have been reported in a study by Venn and Arber [55] suggesting that the inactive retirees are seen as the 'villains of old age'.

> I get mad when I see older people just sitting around, and sometimes I scold them. When they sit there it's hard to get them up again. (#4)

Or

> We never watch the tele during the day. We have other stuff to do and can't just sit there and stare. It's an evening-thing, for relaxation. (#10)

As such, the everyday rhythms are embedded with chrononormativity [56]. Participants are afraid to show idleness to neighbours through television lighting in the windows during daytime or even excessive time at home during the day, as this is seen to portray an inactive lifestyle. The chrononormativity entails specific time for specific activities and allows the participants to become 'exemplars of retirement' [29] by leading active lives and portraying these to the surroundings. While retirees from other countries or with different work-lives and lifestyles might have engaged with the chrononormativity in other ways—e.g., by staying up late, getting up late in the morning, or not having fixed schedules—these participants enact a rather strict norm for how to appropriately manage time. The chrononormativity also extends beyond the daily rhythm, as participants express ideal times for seeing family during the week, and ideal times for being on the boat or golfing during the year. As such, the chrononormativity also relates to social difference and distinction.

The busyness and necessity to be physically active is also a challenge. On the one hand, participants express annoyance about an ageist society expecting little of them, as they feel they are still able to contribute to the labour market. On the other hand, they also feel some of the maladies of old age come sneaking, as they experience their functional capacity decrease and an increase in diseases as well as injuries related to their physical activities. Participants who have 'just had surgery' (#7) or are 'currently unable to do my exercise routine' (#11) due to small or severe injuries are numerous in the follow-up interviews after the RCT. These injuries and diseases are seen as mainly impeding them from further activities and temporarily interrupting their exercise routines. Thus, they expect to be back exercising and continuing their busy lives once they have recovered from their injury or disease.

## 4. Conclusions

While the rhythms we have discussed in this section in some ways are individual choices—after all, who but yourself tells you to set the alarm clock when you are retired—we would rather suggest that the rhythms we have shown are examples of the way retirees continue to engage and participate in society and to perform a busy ethics; this comes with duties. On the one hand, the rhythms are a straightforward necessity. If they want to continue picking up grandchildren, shopping, talking to authorities, GP appointments, etc., they need to follow certain rhythms. However, on the other hand, the activities they engage in, the grandchildren they pick up, the volunteering they provide, and so forth are both a choice to continue societal engagement and a social demand to not become one of the sitting, napping, and daytime tele-watching 'villains of old age' [28]. Therefore, the rhythms are both a societal necessity and a way to live up to the many expectations that come with being a healthy and active senior citizen. They are a way to manage time, and hence, they are a way to manage retirement life.

In this analysis, we have shown how the everyday lives of participants are interwoven by socio/bio/cultural rhythms by using a rhythmanalytical perspective. Rhythmanalysis is a way to study the everyday and the way this is traversed by different biological, social, and regulatory rhythms [57]. With such an approach, the rhythms in our data extend beyond the 13 participants and suggest that retirees are part of society, not just through presence and common human biology, but also through participating in its rhythm.

## 5. Data Limitations

By focusing on the way Danish retirees organise everyday life a cyclical and recurrent notion of time appeared in our data. In our study approach, time appeared implicitly to be linear; we asked

respondents to draw 'straight' timelines of their days' tasks and the ActivPal pedometer recorded number of steps as a proxy measure of physical activity on a linear timeline. However, in the qualitative interviews and focus groups the same linearity of rhythm appeared. As we have shown, the activity monitoring differed from the qualitative data in terms of how participants accounted for differences between weekdays and weekends. However, as the activity monitoring merely measures activity through motion, and not the kind of activity, there can very well be differences in the kinds of activity the participants engage in between weekdays and weekends. Therefore, we would suggest that monitoring activity through accelerometers needs to be supplemented with qualitative methods, which we have done in this study. As a result of this, we have managed to differentiate between weekday and weekend activities.

In addition, it should be noted that when measuring steps—as well as in other studies measuring food intake or other kinds of behaviour embedded with high levels of normativity—participants might alter their behaviour when measured in order to live up to certain norms and standards about activity levels. While measuring behaviour implies a different access to 'actual' behaviour, which we cannot access through the qualitative methods used in this study, the 'actual' behaviour is always altered by participants' awareness of their participation. Participants always negotiate both behaviour, drawing of daily schedules as well as accounts in relation to norms and standards. While they do so in most spheres of life, the adherence to certain norms is likely to be reinforced when participating in a study.

In addition, our analysis of time and rhythms is an example of the ways the questions being asked by the sciences, including the social sciences—in this case through a linear time scale—implicitly enacts a particular version of reality [58]. Here, time is manageable and, as such, becomes an organizing principle in participants' everyday lives. However, the different methods applied in the study creates a framework where other notions of time appear in our data. As we see it, this is exactly why we need to be more creative and interdisciplinary in the ways we craft our research questions and design interdisciplinary research collaborations. By questioning the problematics of time management in retirement with various methods, the different facets of rhythm appear. Moreover, the continuity of rhythms in life pre- and post-retirement, which we find in our data, is the result of an analysis, where we have only been able to study the rhythms pre-retirement retrospectively. We have no 'objective' data that enable us to study the rhythms pre-retirement.

Another general limitation—in this instance about RCTs—should also be mentioned. When conducting research projects on this age group in general (and in particular with this type of RCTs), it is our impression, based on the typical inclusion criteria, that the research participants involved tend to belong to the group of active, engaged, and healthy older people living in or close to university cities. As such, much of the medical, biological, sociological, and psychological knowledge we have on older people is produced with this bias towards people actively volunteering. In our case, we have been transparent about this bias and used it to describe a particular part of the retirement population.

The participants were well-educated, active, and healthy. Hence, these data are not representative for the older population in general but gives insights into the ways this particular group routinize activities and time. However, our rhythmanalytical approach and use of gerontological theory has enabled us to theorize about the daily rhythms of contemporary retirement, which extends beyond the 13 participants and beyond Denmark. Moreover, in Westernized societies in general and in the Scandinavian countries in particular, this well-off group of ageing adults constitute a large part of the older population. Thus, while our findings are locally rooted, the rhythms of contemporary retirement are not just formed locally, but also through societal rhythms.

## 6. Implications

Firstly, it is evident that in order to engage retirees in physical activity, offering activities during the morning hours on weekdays would seem the most feasible and resulting in the best adherence. In general, understanding the rhythms of retirement enables people facilitating activities and initiatives for retirees to design their activity offerings at specific times during the day, week, and season. Likewise,

many participants were annoyed that the long summer breaks at many of their organised activities resulted in loss of physical condition and urged organisers to shorten summer and winter breaks.

Secondly, we would highlight a conceptual implication of our study: Through their daily rhythms, retirees are part of contemporary society and continue to engage and participate in the societal organization. Creating conceptual and spatial boundaries between life pre- and post-retirement neglects the many ways retirement is part of 21st century societies.

**Author Contributions:** Conceptualization, all authors.; methodology, all authors; validation, A.J.L. and K.M.; formal analysis, A.J.L. and K.M.; investigation, all authors.; resources, L.H. and A.P.J.; data curation, all authors; writing—original draft preparation, A.J.L.; writing—review and editing, all authors; visualization, A.J.L. and K.M.; supervision, L.H. and A.P.J.; project administration, L.H. and A.P.J.; funding acquisition, L.H. and A.P.J. All authors have read and agreed to the published version of the manuscript.

**Funding:** CALM was funded by the UCPH Excellence Programme for Interdisciplinary Research.

**Conflicts of Interest:** The authors declare no conflict of interest.

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
