# Peer review of "Retirement Rhythms: Retirees’ Management of Time and Activities in Denmark"

_societies, doi:10.3390/soc10030068_

Round 1

Reviewer 1 Report

I wish to thank the authors for taking the time and patience in addressing my initial critiques of their work. I do feel as if most of my concerns were adequately addressed, and the manuscript overall seems more clear in outlining its objectives and describing its methods.

My only significant concern, based on the edits, is that not enough information is provided regarding participant inclusion/exclusion – I realize the authors refer to another source for this information, but since its blinded for review I essentially have to take everyone’s word for it that the information is there (I understand, it’s just something difficult to get around).

One thing I’ll say though is that, the manuscript needs to be reviewed for grammatical mistakes. I do not recall seeing many in the initial version of the manuscript, but the new version seemed to have more noticeable errors. A few read-throughs to clean up these mistakes is warranted.

Author Response

Many thanks for the review. 

We have now revised the manuscript for grammatical mistakes.

We are aware that this situation regarding the participant inclusion/exclusion is difficult. As will be visible in the references paper, this substudy is part of a larger RCT, in which the recruitment and inclusion process was long and thorough. 

We hope that with the changes made so far this is sufficiently clear. 

Reviewer 2 Report

Authors have incorporated suggestions from Referees

This manuscript is a resubmission of an earlier submission. The following is a list of the peer review reports and author responses from that submission.

Round 1

Reviewer 1 Report

Referee report for Societies, Manuscript – 876265

Retirement Rhythms: Retirees’ Management of Time and Activities in Denmark

The paper is concerned the relevant topic of analysing the management of time and activities of healthy and well-educated pensioners in Denmark.

Authors have combined qualitative (individual interviews, focus group interviews, design games and drawings) and quantitative (activity monitoring, sleep quality and health markers) data from participants over the age of 65 years being research participants in a RCT.

The paper is certainly of great interest for the readers of Societies, since it contains some original work and it is well written.

Author Response

Many thanks for this review. 

Based on the remarks given, we have found no need to further editions. 

Reviewer 2 Report

This study examines retirement among older Denmark residents. Utilizing a combination of quantitative and qualitative data, the authors attempt to examine how the activities engaged in (and the timing of these activities, i.e., a “rhythm”) are reflective of norms experienced prior to retirement. While I think this is an interesting approach (using a theory I was not particularly familiar with), I have a number of questions and issues with the manuscript which I feel require addressing (detailed below):

  • Given how the authors attempt to apply rhythmanalysis to map out the activities of older adults who have retired (a theoretical approach that, while similar to continuity theory, is something I have not seen used in depth in gerontology), I think it would benefit the reader to include a conceptual model of sorts – by illustrating the theory and applicable variables, this may help orient the reader (it also may help give focus on what the objectives of the study are).
  • While I understand that certain project details were omitted to ensure a blinded review, some censured details left me confused about the study. As an example, part 1 of the study (described in the first paragraph under Methods and Materials, pg. 3) is described as an intervention study involving a series of tests and ethnological observations – however, this description does not indicate what intervention was being conducted. Was is educational in nature (e.g., teaching participants healthy exercise routines, healthy eating habits, etc.), or did participants engage in a more active program (e.g., participate in exercise classes)? What was the focus of the intervention? By providing these details, we can get a better sense of what sort of data was ultimately collected.
  • Related to the point above, the description of part 2 of the study (pg. 4 after Figure 1) left me confused in that without a thorough description of part 1, I don’t know what part 2 is assessing when it comes to “knowledge testing.” I also think “innovation project” seems somewhat vague. I think my main issue is that based on the description of parts 1 and 2, I am having trouble understanding how these parts relate to the research questions posed in the introduction.
  • I did not see any explicit recruitment information regarding the participants, nor did I see any explicit inclusion or exclusion criteria (e.g., how were participants recruited? Where were they recruited from? Were they compensated for their participation? Was there screening beyond age, like for cognitive impairment? Etc.). This needs to be added.
  • The description of the qualitative screening and follow-up interviews needs more detail (e.g., what sorts of questions were asked, or what was measured/assessed?).
  • The description of the focus group interviews confused me. If I’m reading it correctly, there were 6 “experimental workshops” (3 on eating, 3 on physical activity), but only 2 (1 eating, 1 physical activity) had participants from the 208 participants? Is that correct (as the description says only 1 from “each theme” had RCT participants)? If not, this needs clarification. And if 1 from each theme had RCT participants, why was only 1 workshop used (why was the eating workshop not used)?
  • In the qualitative coding description (pg. 5), how many researchers were involved in the coding, and how were duties distributed (as an example, were 2 researchers involved who reviewed the coding independently, then met to discuss where they differed)? More information is needed here.
  • Later in the paper it’s clear that sociodemographic data is collected on the participants (sex/gender, education, employment, etc.). This needs to be specified in the quantitative data description.
  • In reviewing the quantitative data, it seems there is some valuable health-related measurements. However, based on the description of the qualitative data, and based on the introduction, I’m having trouble envisioning how this all fits together. What seems to be missing if a very clear and distinct research question with hypotheses to guide the analysis. As I read everything, my big question was: what are the authors trying to answer, and does the data they chose to use actually address it?
  • With only 13 participants, it’s difficult to conduct any robust quantitative analysis (and thus hard to connect any results with the qualitative observations). In addition, because the sample was relatively homogeneous, I worry that we’re just getting a snapshot of older retirees (this authors do concede this, but I feel the need to reiterate it in light of the few number of participants included).
  • A significant limitation to this study is that the authors contend that the rhythms observed in the participants is an extension (or, maybe I should say, continuation/replication?) of the rhythms they experienced when working. While we can assume this to an extent, we have no concrete data to measure what their rhythms were prior to retirement – drawing any conclusions about continuity thus need to be taken with a huge grain of salt (and this should be emphasized in the limitations).
